# Neuromuscular Fatigue in Cerebral Palsy Football Players after a Competitive Match According to Sport Classification and Playing Position

**DOI:** 10.3390/ijerph19106070

**Published:** 2022-05-17

**Authors:** Matías Henríquez, Luis Felipe Castelli de Campos, Fernando Muñoz-Hinrichsen, María Isabel Cornejo, Javier Yanci, Raul Reina

**Affiliations:** 1Sport Research Centre, Department of Sports Sciences, Miguel Hernández University, 03202 Elche, Spain; maria.cornejo@goumh.umh.es (M.I.C.); rreina@umh.es (R.R.); 2Department of Education Sciences, Faculty of Education and Humanities, University of Bío-Bío, Chillan 3780000, Chile; lcastelli@ubiobio.cl; 3Department of Kinesiology, Metropolitan University of Sciences Education, Santiago 7500000, Chile; fernando.munoz_h@umce.cl; 4Society, Sports and Physical Exercise Research Group (GIKAFIT), Physical Education and Sport Department, Faculty of Education and Sport, University of the Basque Country (UPV/EHU), 01007 Vitoria-Gasteiz, Spain; javier.yanci@ehu.eus

**Keywords:** para-football, physical capacity, brain injury, vertical jump, Paralympic

## Abstract

This study aimed to determine the rated perceived exertion (RPE) and match load (RPE-ML) to compare pre-post-match vertical jump (VJ) capacity according to cerebral palsy (CP) players’ sport classes (i.e., FT1–FT3) and playing positions and to explore whether the neuromuscular performance variation is associated with the internal load of para-footballers with CP. Fifty-six male para-footballers performed two VJ tests before and immediately after a competitive CP football match, followed by measurements of the players’ RPE and RPE-ML. There were no significant differences (*p* > 0.05) in the pairwise comparisons for RPE and RPE-ML according to sport classes and playing position. A significant reduction in the VJ performance was found for each player sport class and playing position in squat jump (SJ) (*p* < 0.01; 0.24 < *d_g_* < 0.58) and countermovement jump (CMJ) (*p* < 0.05; 0.22 < *d_g_* < 0.45). Regarding the pairwise comparisons, players with the minimal impairment criteria (FT3) obtained higher deficit scores during SJ than those belonging to the FT1 and FT2 (*p* = 0.003; 1.00 < *d_g_* < 1.56). Defenders experienced the lowest performance compared to midfielders and attackers in SJ performance (*p* = 0.027; 0.94 < *d_g_* < 1.28). Significant correlations were obtained between ΔSJ or ΔCMJ and RPE or RPE-ML (*r* = −0.58 to −0.75; *p* < 0.001). These findings provide novel information supporting the notion that fatigue induced after a competitive match causes notable impairments in VJ performance differentiated according to sport class and playing position in para-footballers with CP.

## 1. Introduction

Cerebral palsy (CP) football is a team para-sport that is played 7-a-side on a 70 m × 50 m pitch (goals 5 m × 2 m), where participants must present a permanent neurological impairment of hypertonia, ataxia, or athetosis [1]. These body function deficiencies are provoked by congenital CP or related underlying health conditions (e.g., acquired brain injury), affecting the performance of motor activities [2]. In addition, CP is described as the most common motor disability of childhood, and it is considered to encompass a group of conditions with variable severity, often accompanied by secondary disturbances in different body systems and functions such as balance, coordination, or muscle tone [2]. Currently, CP football classification comprises three sports classes according to the impact of the eligible impairments on general and football-specific skills performance [3]. Hence, para-footballers are classified as FT1, FT2, or FT3 according to the severity of the activity limitation (i.e., from more to less severe impact), but also considering their functional profile [i.e., bilateral spasticity or diplegia, overall coordination impairment (i.e., athetosis or ataxia), and unilateral spasticity or hemiplegia]. The sport class allocation is assigned after physical, technical, and observational assessments prior to the competition [1]. Overall, people with CP present the following: compromise of the descending neural pathways that control body movement, recruitment of motor units, produce muscle tone alterations, co-contraction, and progressive secondary musculoskeletal complications over the lifetime [2]. Specifically, CP football players (CPFPs) present neuromuscular compromise affecting endurance capacity [4], muscular strength [5], change of direction ability [6], repeated sprint ability [7], acceleration and deceleration performance [8], jump capacity [9], and whole physical performance during training or competition instances [3,10]. Given that CP football requires multiple actions of very high intensity and high neuromuscular capacity [7,11], which must be maintained throughout the competitive actions, it may be relevant to know the muscle fatigue of para-footballers induced by matches. There are several methods to quantify fatigue in football [12,13]; despite this, one of the most used is vertical jump (VJ) variation (pre- to post-match performance difference) analysis as an indirect neuromuscular fatigue indicator induced from match play [14,15,16,17]. Previous studies suggest that changes in VJ performance following football matches are related to neuromuscular fatigue due to the specific demands and the requirements of the anaerobic capacity in the lower limbs [18,19,20]. In this sense, several studies that used VJ assessment, such as squat jump (SJ) and countermovement jump (CMJ) to monitor simulated or real post-match fatigue, demonstrated declines in height jump performance, using these fast and easy procedures [15,17,18,21].

Although SJ and CMJ protocols have been used and validated with CPFPs [9,22], to the best of the authors’ knowledge, no previous study has analyzed the use of these tests to describe the impact of match demands on neuromuscular fatigue in this para-sport. Moreover, considering the differences of the physical capacity of CPFPs according to their sport classes [11,23], it could be pertinent to know whether the fatigue presents a similar magnitude of impact in the different player functional profiles. On the other hand, even in regular football [24,25], fatigue could be conditioned according to the playing position and the response activity profile to match demands. With regard to this variable, no previous studies have analyzed how the players’ playing position influences the neuromuscular fatigue of CPFPs. Knowing the monitorization of post-match fatigue by using quick and simple methods, such as jump assessment, in para-footballers with CP could be essential to managing the recovery-process strategies planning training load sessions and identifying possible individual impairment-profile differences in fatigue-related responses [12]. 

Therefore, this study aimed (1) to determine the perceived match load according to sport classes and playing position, (2) to compare neuromuscular match-induced fatigue (i.e., pre-post-match jumping capacity variation) according to players’ sport classes and playing positions, and (3) to explore whether the neuromuscular performance variation is associated with the perceived fatigue and match load of CPFPs.

## 2. Materials and Methods

### 2.1. Participants

A convenience sample of 56 male CPFPs (25.0 ± 7.0 years, 68.7 ± 11.7 kg, 1.70 ± 0.07 m, 23.9 ± 4.1 kg·m^−2^) participated in this study. All the players competed in the Chilean CP football league, and 15 of them belong to the national team. The participants’ sport classes are described according to the classification rulebook of the International Federation of Cerebral Palsy Football (IFCPF) [1] and their regular playing position (i.e., defender (DEF), midfielders (MF), and attacker (AT)) (Table 1). The inclusion criterion for the study was to complete more than 50 min of the entire match playing time (i.e., official match time 60 min). Goalkeepers were excluded from this study due to the specialization of their position. All the players were given detailed verbal instructions and were fully familiarized with the procedures before testing. The study’s characteristics were detailed in an oral explanation to the participants, and their written informed consent was obtained in concordance with the declaration of Helsinki (2013). The Human Ethics Committee of the Santo Tomas University (reference no. 63.20) approved all the procedures of this study.

### 2.2. Procedures

All the participants belonged to CP football club teams, which competed in a local tournament held in Chile. Six competitive matches were played on different days on the same field of normal size and in concordance with the official requirements of the IFCPF [26]. The matches consisted of 14 players (2 goalkeepers and 12 outfield players) and were played in a regular competition environment. All matches were performed outdoors on a synthetic grass pitch, with an ambient temperature between 20 and 23 °C and a relative humidity of 54.2 ± 5.2%. The perceived load exertion was assessed in the players following each competitive match. In order to determine neuromuscular fatigue as conducted in previous studies with football players [14,15], the participants performed two VJ tests 30 min before starting the match during the team warm-up and immediately after the competitive CP football matches.

#### 2.2.1. Match Intensity and Load

The rated perceived exertion (RPE) has increasingly been used to measure internal load and to determine the perceived effort in non-impaired football players [13,24,25,27] and previously utilized in players with CP [10,28]. This method has been described as being valid for quantifying efforts during training and football matches [29,30,31] and has been proposed for use in the context of football and other team sports [32,33]. For example, Foster et al. [34,35,36] proposed the RPE to assess the “hardness” of the entire training session and to evaluate internal load in endurance and team-sport athletes. In addition, the RPE provides information in a simple, practical, and low-cost way [37]. Especially in high-level performance contexts, where the use of technology is constrained due to the discomfort when players are at peak performance in competition, the RPE can be an excellent non-invasive method applied once the match is over [36], minimizing the intrusion in the collection of information. Due to the validity of this method and its facility, versatility, and low cost, the RPE was used in this study to quantify the internal load of the players during the matches. All the participants responded with regard to their perception on a 0-to-10-point scale [36], following 30 min of the official match to ensure that the perceived effort was referred to the whole session [31]. The players responded separately, without the presence of other players, to the RPE scale, which was administered by the same two persons (i.e., author researchers) [15,24]. Additionally, the internal match load was calculated by using the proposed method by Foster et al. [36] and Impellizzeri et al. [31], as follows: [perceived match load (RPE-ML) = RPE value × match time (min)], expressed in arbitrary units (AU). The match time was considered, excluding warm-up and rest periods [38]. This match load quantification method was also used in a previous study with CPFPs [28].

#### 2.2.2. Neuromuscular Capacity

The neuromuscular capacity was analyzed by assessing VJ performance [14,17], where SJ and CMJ were considered. The VJ test was performed by using a jump platform (2.5, DMJUMP, Santiago, Chile) on a stable surface and using the same protocol described previously in CPFPs who used the CMJ [9] and the SJ [22]. These tests were selected for a fast and straightforward application assessment, as has been performed in other studies analyzing the influence of match demands on physical parameters [14,17,18,21]. In the SJ, participants were instructed to perform two maximum VJs from an initial position with a knee flexion angle of 90° maintained for 2 s, before jumping without any rebound or countermovement [39]. During the starting to final jump position, the hands stayed on hips, and in the takeoff phase, the participants remained with their legs fully extended. In the landing phase, participants touched down with both feet together in an upright position. With regard to the CMJ, the participants started in a standing position, with fixed hands-on hips and knees fully extended, sustaining this position during the jump execution without using arm swing [22]. From this position, all participants performed a fast flexion-extension downward movement until the knee angle reached 90°, followed by performing a maximum VJ effort with legs fully extended and plantar flexion during the takeoff phase and landing on the starting place. For both tests, the jumps were made in two attempts, with rest intervals of one min between trials, registering the highest height obtained (cm). Those athletes who presented spastic hemiplegia and difficulty maintaining their hands on their hips were allowed to keep their hands at the sides of their body [22]. 

### 2.3. Data Analysis

All results were calculated as mean ± standard deviation (SD). Kolmogorov–Smirnov and Levene’s tests were applied to verify the distribution and homogeneity of data. A mixed 2 × 3 × 3 repeated-measures analysis of variance (ANOVA) was conducted, considering the match (pre-match vs. post-match measurements) as a within-group factor and the CP football sport classes (i.e., FT1, FT2, and FT3) and the playing positions (i.e., DEF, MF, and AT) as the between-group factors. A Tukey’s post hoc analysis was used to examine the pairwise differences among sport classes and playing positions. Student’s paired *t*-test was applied to determine the differences between pre- and post-VJ height performance in each subgroup of players according to their sport class and playing position. The delta value for height jumps’ variation (Δ cm) in SJ and CMJ was calculated by using the following formula: Δ = (post-match jump height) − (pre-match jump height). Two effect-size indexes were used to assess the practical within- and between-group differences. On the one hand, partial eta-square (η_p_^2^) values were calculated as a measure of effect size for mean differences with the following interpretation: above 0.26, between 0.26 and 0.02, and lower than 0.02 were considered as large, medium, and small, respectively [40]. On the other hand, to calculate the effect size of post hoc within-group differences, Hedges’ *g* index was used [41]. This index is based on Cohen’s *d* index [42], but it provides an effect-size estimation, reducing the bias caused by small samples (i.e., subgroups with n < 20). Interpretations of Hedge’s *g* that were above 0.80, between 0.50 and 0.79, between 0.25 and 0.49, and lower than 0.25 were considered large, moderate, small, and trivial, respectively [41]. The pre-post performance differences were also calculated to explore pairwise differences according to sport classes and playing positions with their respective upper and lower confidence intervals. The relationship between VJ variation and perceived match load variables was calculated by using linear Spearman’s correlation (*r*). The correlation coefficients were qualitatively interpreted as follows: <0.09, trivial; 0.10–0.29, small; 0.30–0.49, moderate; 0.50–0.69, large; 0.70–0.89, very large; and >0.90 nearly perfect [43]. Data analyses were performed by using the statistical package GraphPad Prism (GraphPad Software, version 8 for Windows, San Diego, CA, USA) and the Statistical Package for Social Sciences (SPSS Inc., version 26.0 for Windows, Chicago, IL, USA). Statistical significance was set at *p* < 0.05.

## 3. Results

In general, the overall sample results showed a mean RPE-ML of 370 ± 89 AU and RPE of 6.3 ± 1.5. The mean played time of the participants was 58.9 ± 2.1 min. The RPE and RPE-ML responses of the participants among sport classes and playing positions are presented in Figure 1. With regards to the sport classes, the obtained scores were 376 ± 77 AU and 6.4 ± 1.3 for FT1, 365 ± 89 AU and 6.2 ± 1.5 for FT2, and 387 ± 104 AU and 6.4 ± 1.7 for the FT3. In terms of the playing positions, the observed values were 328 ± 93 AU and 5.5 ± 1.6 for the defenders, 370 ± 88.2 AU and 6.3 ± 1.5 for the midfielders, and 390 ± 86 AU and 6.6 ± 1.4 for the attackers. No significant differences were found for the overall and pairwise comparisons for the RPE and RPE-ML variables according to the two between-group factors of this study (i.e., sport classes and playing position).

For the overall sample, the VJ height was reduced following the football match, both in the SJ [23.7 ± 6.5 cm to 21.0 ± 5.7 cm, Δ = −2.71 ± 1.55 cm: F(1,48) = 93.46; *p* < 0.001; η_p_^2^ = 0.88, large] and CMJ [24.4 ± 6.7 cm to 21.6 cm ± 5.9 cm, Δ = −2.81 ± 1.87 cm: F(1,48) = 44.17; *p* < 0.001; η_p_^2^ = 0.48, large] measurements. No significant differences in the pretest or in the post-test were found in VJ performance between players’ sport classes (*p* = 0.064–0.216) and playing position (*p* = 0.381–0.765) in each period of pre- and post-match measurements. Interaction effects between the within-group and between-group factors were found only for the pre–post measurement of the SJ and the sport classes [F(2,48) = 4.50; *p* = 0.016; η_p_^2^ = 0.16, medium]. Figure 2A,B shows the para-footballers’ VJ performance according to their sport classes, while Figure 3A,B illustrates them in terms of considering their playing positions. With respect to the player’s sport class, the paired *t*-test indicated a significant difference in VJ performance between pre–post-match measurements in all the classes for SJ (Figure 2A) in FT1 (*t*(6) = 6.20; *p* = 0.001, *d_g_* = 0.30, small), FT2 (*t*(39) = 11.37; *p* < 0.001, *d_g_* = 0.42, small), FT3 (*t*(8) = 7.05; *p* < 0.001, *d_g_* = 0.58, moderate), and CMJ (Figure 2B) in FT1 (*t*(6) = 2.69; *p* = 0.036, *d_g_* = 0.31, small), FT2 (*t*(39) = 9.51; *p* < 0.001, *d_g_* = 0.45, small), and FT3 (*t*(8) = 5.82; *p* < 0.001, *d_g_* = 0.45, small). 

With regards to playing position, after the matches significant differences were found for the SJ (Figure 3A) performance for DEF (*t*(8) = 4.36; *p* = 0.002, *d_g_* = 0.24, trivial), MF (*t*(27) = 9.27; *p* < 0.001, *d_g_* = 0.47, small), and AT (*t*(18) = 9.75; *p* < 0.001, *d_g_* = 0.42, small). These differences were also found for CMJ (Figure 3B) in the playing positions of DEF (*t*(8) = 2.42; *p* = 0.042, *d_g_* = 0.22, trivial), MF (*t*(27) = 7.93; *p* < 0.001, *d_g_* = 0.48, small), and AT (*t*(18) = 9.50; *p* < 0.001, *d_g_* = 0.44, small).

Table 2 shows the VJ performance decrement and pairwise comparison after the match in relation to CP football sport classes and playing positions. With regard to sport classes, significant differences were found for the SJ performance deterioration. Observing the pairwise comparisons, players with the minimal impairment criteria (FT3) obtained higher deficit scores during SJ capacity than those belonging to the FT1 and FT2 sport classes. Table 2 also reports the VJ deficits according to the playing positions. After the matches, significant differences were also found for the SJ performance deterioration, where defenders experienced the lowest performance compared to midfielders and attackers. 

Looking at the associations between the ΔVJ performance with RPE and RPE-ML, we see that large inverse correlations (*p* < 0.001) were obtained between ΔSJ and RPE (*r* = −0.59) and RPE-ML (*r* = −0.58). Similar results were obtained between ΔCMJ and RPE (*r* = −0.75, *p* < 0.001) and RPE-ML (*r* = −0.74, *p* < 0.001).

## 4. Discussion

The purpose of this study was to compare the RPE and RPE-ML in football matches reported by CPFPs of different sport classes and playing positions to determine the impact of a competitive CP football match on the VJ performance variation as a neuromuscular fatigue indicator, also taking into account the players’ sport classes and their playing positions, and to describe the association between VJ height variation and the RPE-ML or RPE. The major findings were as follows: (1) no differences were obtained for RPE-ML and RPE between sport classes and playing position; (2) CPFPs have significant decrements in SJ and CMJ performance across sport classes and playing positions after a competitive CP football match, evidencing an altered neuromuscular performance that affects jumping capacity; (3) during the SJ, FT3 had a higher jumping deterioration than FT1 and FT2 following the match, and MF and AT had a higher jumping deterioration than DEF; and (4) significant associations were found between VJ test decrements and RPE or RPE-ML.

The present results showed no significant differences between sport classes in the RPE-ML and RPE. In this regard, Henríquez et al. [10] found no differences between the sport classes in the RPE and the heart rate of CPFPs following the performance of two small-sided games and a simulation match, reinforcing the possibility that the mentioned factors may explain the absence of differences in the internal load across functional profiles. Contrary to the obtained results, previous studies observed differences in the external load of players’ sport classes during international matches, showing the influence of the impairment profiles on physical response parameters [3,11]. Possibly, due to the functional differences between classes and despite the lower external load of FT1 and FT2 (more affected classes) than FT3, the internal load or the perceived exertion could be similar following a match. These results show that the player sport classes with more impairment, despite having a lower physical response than the more functional profiles, have a similar perceived perception effort [10]. Therefore, it would be interesting to quantify in CPFPs not only the external match response but also the internal or perceived load, because this could provide complementary match load information. For future studies, to confirm whether the internal load is similar, it would be necessary to quantify this by using methods such as heart rate monitoring in competitive conditions [16]. With regard to the playing positions, no differences were found in the perceived load variables in the competitive match. These findings are consistent with what has been found in previous studies during matches with nondisabled professional football players [44,45]. However, other studies reported differences according to the perceived load exertion and playing position during competitive matches [24,25,46]. Nevertheless, the para-sport specific particularities must be considered in the analysis, where the playing positions are not entirely delimited, there is no offside rule, there is a presence of a lower number of players (7-a side), and there is a smaller field size than regular football, factors which could influence tactical strategies and the perceived exertion load during matches [11,47]. Even though the present study did not find differences in the RPE and ML according to sports classes and player positions, the use of these methods could be helpful in a practical setting to perform a post-match individual monitorization considering the specific CP football characteristics.

After a competitive match of CP football, the overall sample results indicated a decrease in VJ performance (SJ and CMJ). Similar to these results, a decrease in the VJ performance was also demonstrated after football matches with nondisabled footballers [12,14,16,17] and in a group of amputee para-footballers [48]. The competitive demands of CP football matches require high-intensity activities involving short-term actions by players who present specific impairments related to neurological conditions [11,47]. These high neuromuscular demands and the necessity of maintaining a work rate over the competitive match time in an intermittent activity could be one cause of the development of neuromuscular fatigue [49]. In addition, given that CP athletes have reported lower levels of muscle strength [5], asymmetrical adaptations for jumping tasks [50], and pacing strategies to compensate fatigue components during motor activities [51], these factors may contribute to greater VJ decrements after football-match activities.

With regard to each sport class result, a significant reduction in the VJ height was found across the functional profiles, showing neuromuscular-induced fatigue following the competitive match on SJ and CMJ measurements. Furthermore, regarding the pairwise comparison, FT3 players presented a higher marked reduction in SJ performance after the matches than their FT2 and FT1 counterparts. Players with the minimal impairment criteria (FT3) are described as performing best in very high-intensity activities and exhibit the best motor performance compared to other sport classes [3,9]. Considering that only one FT3 player per team can play at the same time and that this is the profile with the greater functionality [26], these players may have higher physical requirements during the game, a factor which would produce more muscle damage and fatigue, affecting their capacity of force production post-match [14]. However, regarding CMJ, no significant differences were obtained between sport classes. The CMJ is a widely used method for the assessment of fatigue; however, the neuromuscular aspects related to the fatigue can also manifest as an alternative movement strategy, variations of kinetic variables, and, in this case, possibly the influence of the functional profile [52]. Accordingly, Reina et al. [9] found that horizontal jumps (i.e., standing broad jump, four bounds for distance, and triple hope for distance) presented more differences between sport classes than VJ, based on each jump’s specific demands (i.e., higher demands of coordination and balance). The non-significant differences between sport classes in CMJ could reflect an individual impairment-specific response to the match demands, where concentric force-generation was more affected and reflected during SJ performance [21,53]. Furthermore, according to Van Hooren and Zolotarjova [54], individuals with an impaired coordination function or inadequate capacity to correctly time muscle activation could have worse performance in SJ; however, they could perform a CMJ relatively well due to the capacity to uptake muscle slack and the buildup of stimulation during the countermovement. Thus, it is plausible to suggest that, for SJ, the height decrements were more marked according to the different classes, though the impact of fatigue expressed in CMJ performance had a similar influence across CP football profiles.

When examining the playing positions, a significant decrease was shown for SJ and CMJ among all position groups after the competitive match. For SJ, in comparing playing positions, a lower significant deterioration was reported in defenders than midfielders and attackers. In this regard, Mohr et al. [55] showed that non-disabled footballers with a role as midfielders and attackers performed more high-intensity activities (i.e., the distance at a high intensity) than defenders, and they experienced fatigue toward the end of matches, as well as temporarily during the game, independent of playing position. Similarly, previous studies reported that, during professional matches, central defenders performed a lower distance covered at different intensities compared to other playing positions [56,57]. Indeed, multiple factors could affect the work rate and produce fatigue during the match, such as the tactical role [58], level of play [59], and physical fitness [49]. Unfortunately, in the case of CPFPs, as far as the authors are aware, no studies have analyzed whether the match load is different for players depending on the playing position. Caution is necessary to interpret the findings of this study due to the different functional performance based on the disability profile in the case of para-athletes with CP and considering the CP football’s technical/tactical particularities, which can differ from regular football. The significant performance loss found in VJ after a competitive match should concern coaches for addressing adequate recovery periods during congested calendars or after intense games, especially for those who showed more jump height decrements, such as FT3 players and those with MF or AT playing roles.

The relationship found in the present study between VJ deterioration and perceived load parameters indicates that players with lower perceived exertion during the match presented lower reductions in VJ height. Rampinini et al. [60] reported significant relationships between player match RPE and peripheral fatigue indicators related to a reduced muscle contractile capacity. Contrary to these findings, Benítez-Jiménez et al. [58] showed no significant relationship between the change in CMJ height and the average RPE following consecutive friendly matches in young football players. It seems that those CPFPs with higher RPE or RPE-ML values would present greater neuromuscular fatigue represented by a VJ deterioration following a competitive match. Therefore, from a training point of view, it could be relevant for coaches to implement training protocols oriented to obtaining a superior physical fitness to promote a reduction in the match perceived exertion, without a reduction in physical requirements, and consequently produce less neuromuscular fatigue [13,46]. These findings provide novel information regarding relationships between perceived load and neuromuscular fatigue produced in CPFPs after competitive matches, approaches which could improve the physical assessment of this group of para-footballers.

Although this research has been carried out with high scientific standards and methodological rigor, the main limitation is that objective methods have not been used to quantify match load. Taking into account the possible limitations that the RPE-ML may have, it would be interesting if, in future studies, the quantification of the match load could be complemented with objective load quantification methods.

## 5. Conclusions

The VJ testing could provide data for coaches’ training sessions, the implementation of recovery strategies, and the assessment of neuromuscular fatigue of para-footballers with CP. The SJ and CMJ were demonstrated to be useful in assessing neuromuscular fatigue in CP footballers; however, the SJ should be considered in terms of the characteristics of each player’s sports classes and playing positions. These results should be interpreted with caution due to the number of participants in each group, according to sport classes and playing positions. It is also necessary to consider the influence of physical, technical, and tactical demands specific to different classification profiles, and the absence of the assessment for additional neuromuscular fatigue variables (e.g., repeated sprint ability, sprint, biochemical markers, maximal voluntary strength, or kinematic markers). The present data provide novel information suggesting that VJ tests could be a useful indirect measure of neuromuscular fatigue expressed in height jump loss after a football match, highlighting that CP football can be a modality with high neuromuscular demands. The SJ performance pointed out some differences regarding fatigue manifestation, considering players’ sport classes and playing positions, especially for less impaired players (i.e., FT3) and those assuming tactical roles in midfield and forward positions. A practical application of using VJ tests is that technical staff could indirectly estimate neuromuscular and fatigue status, providing necessary information to adjust individual recovery strategies according to CP footballers’ requirements [14]. A possible limitation of the present study is that it did not include other internal or external load variables, as this, in conjunction with the use of subjective perception tool and VJ, could provide a better comprehension of the load-fatigue process in competitive matches of CP football. Further studies could also consider larger international-level sample sizes to compare sports performance according to playing position and/or sport classes, in combination with the addition of multiple variables that are relevant to neuromuscular fatigue and the time-course recovery that can allow the design of individualized training programs approaching the competitive season. Moreover, more research is needed to explore the use of methods differentiating between respiratory and muscular perceived effort, alternatives that could provide an individual approach to quantify the internal load in para-footballers with CP [38]. In agreement, future research would examine differences in the match RPE between CP football players and non-impaired footballers, hypothesizing that data may provide insights to understand the feasibility of the use of tools for the subjective perception of the efforts in the specific para-sport contexts [51].

## Figures and Tables

**Figure 1 ijerph-19-06070-f001:**
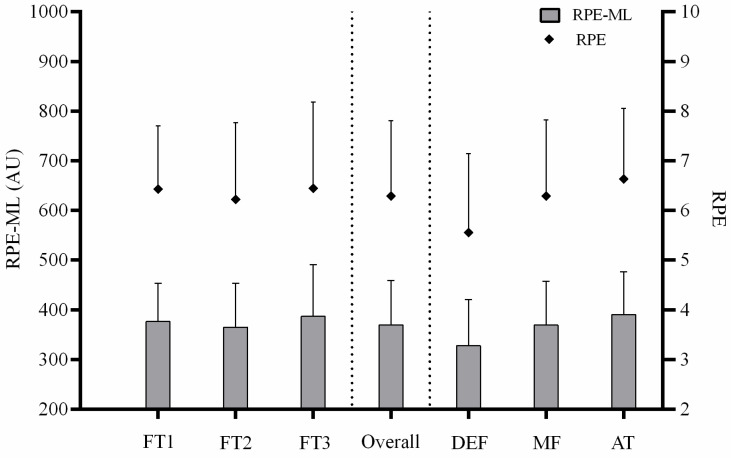
Perceived match load (RPE-ML) and perceived exertion (RPE) of participants among sport classification and playing position. FT1–FT3, cerebral palsy football sport classes; DEF, defender; MF, midfielder; AT, attacker.

**Figure 2 ijerph-19-06070-f002:**
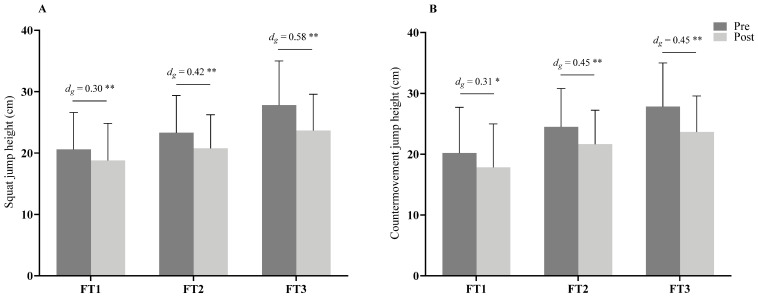
Vertical-jump height performance following football match on squat jump (**A**), and countermovement jump (**B**) among sport classes. FT1–FT3, cerebral palsy football sport classes, *d_g_* = effect size; ** (*p* < 0.01) and * (*p* < 0.05) are significant difference between pre- and post-match.

**Figure 3 ijerph-19-06070-f003:**
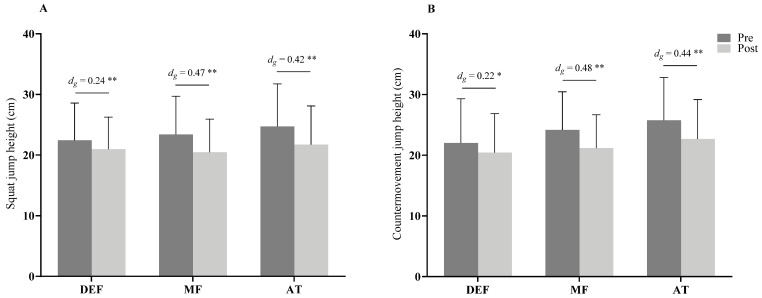
Vertical jump height performance following football match on squat jump (**A**) and countermovement jump (**B**) among playing position. DEF, defender; MF, midfielder; AT, attacker, *d_g_* = effect size; ** (*p* < 0.01) and * (*p* < 0.05) are significant difference between pre- and post-match.

**Table 1 ijerph-19-06070-t001:** Cerebral palsy football player’s characterization according to sports classification and playing position.

	FT1	FT2	FT3
Number of players	7 (12.5%)	40 (71.4%)	9 (16.1%)
Impairment			
Bilateral Spasticity	2 (3.6%)	--	--
Coordination Impairments	4 (7.1%)	--	1 (1.8%)
Unilateral Spasticity	1 (1.8%)	40 (71.4%)	8 (14.2%)
Playing Position			
Defender	2 (3.6%)	7 (12.5%)	--
Midfielder	1 (1.8%)	23 (41.1%)	4 (7.1%)
Attacker	4 (7.1%)	10 (17.8%)	5 (8.9%)

FT1–FT3: cerebral palsy football sport class.

**Table 2 ijerph-19-06070-t002:** Vertical jump performance variation (Δ cm, pre-match vs. post-match assessments) according to sport classes and playing position.

	FT1	FT2	FT3	*d_g_* Pairwise Comparisons (LCI–UCI)			*p*
				FT1 vs. FT2	FT1 vs. FT3	FT2 vs. FT3	
ΔSJ (cm)	−1.8 ± 0.8 (−9.4%)	−2.5 ± 1.4 (−10.9%)	−4.2 ± 1.8 (−14.7%)	0.51 (−0.45–1.57)	1.56 ** (0.42–2.70)	1.00 ** (0.02–1.99)	0.003
ΔCMJ (cm)	−2.3 ± 2.3 (−12.1%)	−2.8 ± 1.9 (−11.4%)	−3.1 ± 1.6 (−11.0%)	0.25 (−0.55–1.06)	0.39 (−0.61–1.39)	0.16 (−0.56–0.88)	0.733
	**DEF**	**MF**	**AT**	**DEF vs. MF**	**DEF vs. AT**	**MF vs. AT**	** *p* **
ΔSJ (cm)	−1.4 ± 1.0 (−6.1%)	−2.9 ± 1.7 (−12.4%)	−3.0 ± 1.3 (−12.1%)	0.94 * (0.15–1.72)	1.28 * (0.41–2.14)	0.06 (−0.52–0.64)	0.027
ΔCMJ (cm)	−1.6 ± 2.0 (−6.8%)	−3.0 ± 2.0 (−12.3%)	−3.1 ± 1.4 (−12.3%)	0.68 (−0.08–1.45)	0.91 (0.08–1.74)	0.06 (−0.53–0.64)	0.099

Δ, deficit change; SJ, squat jump; CMJ, countermovement jump; FT1–FT3, cerebral palsy football sport classes; DEF, defender; MF, midfielder; AT, attacker; LCI, lower interval of confidence; UCI, upper interval of confidence. Repeated-measures ANOVA significant differences between FT classes or playing positions: ** *p* < 0.01 and * *p* < 0.05.

## Data Availability

All relevant data are within the paper.

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
