# Peer review of "Neuromuscular Fatigue in Cerebral Palsy Football Players after a Competitive Match According to Sport Classification and Playing Position"

_ijerph, 2022, doi:10.3390/ijerph19106070_

Round 1

Reviewer 1 Report

Congratulations on your work, research on cerebral palsy and physical exercise is commendable.

However, the originality and relevance of the manuscript are limited. 

In the introduction it is recommended to expand the information related to cerebral palsy.

Are there differences in perceived effort and fatigue between conventional football and adapted football?

Is it possible to compare regular football with football adapted to cerebral palsy? Could these premises be considered for future studies?

Author Response

Answers to Reviewer 1

Congratulations on your work, research on cerebral palsy and physical exercise is commendable. However, the originality and relevance of the manuscript are limited.

Author's Answer (AA): We would like to thank the reviewer for the helpful advice and recommendations regarding this manuscript, with ID ijerph-1664873. Concerning the comment on the originality and relevance of the research, no studies have analyzed parameters indicating fatigue in football players with cerebral palsy (CP). Moreover, the present study results may have a direct practical application in this para-sport. We revised the manuscript based on your suggestions on a point-by-point basis. In the following lines, we have included all the comments and clarifications for the reviewer's queries, informing you about the modifications in the manuscript's new version (in red colour).

#1: In the introduction it is recommended to expand the information related to cerebral palsy.

AA#1: Thank you for your suggestion. In addition to the described in the introduction about CP, more information has been included to expand the information on the required topic as follows (Lines 41‒44):

"In addition, CP is described as the most common motor disability of childhood and it is considered a group of conditions with variable severity, often accompanied by secondary disturbances in different body systems and functions such as balance, coordination, or muscle tone [2]."

#2: Are there differences in perceived effort and fatigue between conventional football and adapted football?

AA#2: Thank you for your query. According to the author's knowledge, no study has explored if there are differences in the perceived effort and fatigue between conventional football and adapted football. However, Runciman et al. (2016) compared para-athletes with CP against able-bodied participants realizing a shuttle protocol that considers the rating of perceived exertion (RPE) and other variables, and they demonstrated that para-athletes registered lower RPE values. The available studies in CP football, only have described the characteristics of this specific sample, exploring the differences in the perceived effort performing two formats of small-sided games and a simulation match (Henríquez et al., 2021) and monitoring the internal load of training sessions in Brazilian para-footballers (de Freitas et al., 2020).

References

de Freitas, V.H.; Nakamura, F.Y.; Faria, F.R. de; Dantas, M.J.; Souza, N.C.; Buratti, J.R.; Freitas, A. de; Figueiredo, D.H.; Golçalves, H.R.; Junior, W.N.; Gorla, J.I. Internal training load and performance indices of cerebral palsy football players and effects of one week with and without training on heart rate variability. J. Phys. Educ. Sport. 2020, 20(5), 3017–3022. doi:10.7752/jpes.2020.s5410

Henríquez, M.; Iturricastillo, A.; González-Olguín, A.; Herrera, F.; Riquelme, S.; Reina, R. Time−motion characteristics and physiological responses of para-footballers with cerebral palsy in two small-sided games and a simulated game. Adapt. Phys. Act. Q., 2021, 1–16. doi:10.1123/apaq.2020-0077

Runciman, P.; Tucker, R.; Ferreira, S.; Albertus-Kajee, Y.; Derman, W. Paralympic athletes with cerebral palsy display altered pacing strategies in distance-deceived shuttle running trials. Scand. J. Med. Sci. Sports. 2016; 26(10), 1239–1248. doi:10.1111/sms.12575

#3: Is it possible to compare regular football with football adapted to cerebral palsy? Could these premises be considered for future studies?

AA#3: Thank you for these queries. As mentioned in the second answer, limited research addresses this topic in para-athletes with CP. Regarding your first question, based on the statement that CP football involves similar nature characteristics as regular football, it could be feasible to compare the perceived exertion of CP and non-impaired players. However, at the same time, it will be necessary to consider the sports rules differences that could impact internal player load (i.e., match time, field size, number of players), where for further comparative studies, a protocol that solves these discrepancies are needed (Yanci et al., 2018). In addition, the exploration of differentiating muscular and respiratory rating of perceived exertion in para-footballers is also an unexplored issue. According to this, the following sentence has been included in the conclusions section as follows (Lines 393‒399):

Also, more research is needed to explore the use of methods differentiating between respiratory and muscular perceived effort, alternatives that could provide an individual approach to quantify the internal load in para-footballers with CP [30]. In agreement, future research would examine differences in the match RPE between CP football players and non-impaired footballers, hypothesizing that data may provide insights to understand the feasibility of the use of tools for the subjective perception of the efforts in the specific para-sport contexts [53].”

  1. Runciman, P.; Tucker, R.; Ferreira, S.; Albertus-Kajee, Y.; Derman, W. Paralympic Athletes with Cerebral Palsy Display Altered Pacing Strategies in Distance-Deceived Shuttle Running Trials. Scand. J. Med. Sci. Sport. 2016, 26, 1239–1248, doi:10.1111/sms.12575.

Reference

Yanci, J.; Castillo, D.; Iturricastillo, A.; Urbán, T.: Reina, R. External match loads of footballers with cerebral palsy: A comparison among sport classes. Int. J. Sports Physiol. Perform. 2018, 13(5), 590–596. doi:10.1123/ijspp.2017-0042.

Reviewer 2 Report

Dear authors,

Your article is scientifically sound, detailed and ethical.

great disclaimer at line 370

Future:

Doing this same study with non-CP FPs as comparison would be a good and important visualisation of the difference 

And also doing this same study but with one team of only FT1, 1 team of only FT2 and 1 team of only FT3 ( 1 team per CP football sport class where equal numbers of CPFP from each class have the same playing position (same number of DEF, MF, AT) would also be great.

(noted in the discussion by the authors)

of course a larger sample size,

but the high level of the players is great here (many being in the national team).

the findings are significant and especially the difference between the post match difference in SJ & CMJ vs VJ performance is interesting and great to have published.

great detailed discussion also highlighting findings of previously published material and answering some before the discussion unanswered questions regarding comparison to non cerebral palsy football players performance.

great and valid and important disclaimer again, at line 344

I look forward to seeing your article published.

kind regards.

Author Response

Answers to Reviewer 2

Dear authors, your article is scientifically sound, detailed and ethical.

Author's Answer (AA): We would like to thank the reviewer for the helpful advice and recommendations regarding this manuscript with ID ijerph-1145669. We revised the manuscript based on your suggestions on a point-by-point basis. In the following lines, we have included all the comments and clarifications for the reviewer's queries, informing you about the modifications in the manuscript's new version (in red colour)

#1: great disclaimer at line 370

AA#1: Thank you for this comment. This sentence tries to describe the multiples variables not yet explored in CP football.

#2: Future: Doing this same study with non-CP FPs as comparison would be a good and important visualisation of the difference. And also doing this same study but with one team of only FT1, 1 team of only FT2 and 1 team of only FT3 (1 team per CP football sport class where equal numbers of CPFP from each class have the same playing position (same number of DEF, MF, AT) would also be great. (noted in the discussion by the authors) of course a larger sample size, but the high level of the players is great here (many being in the national team).

AA#2: Thank you for these suggestions. The comments by the reviewer are interesting, while the problem with the proposal is that according to the CP football rules, at least a minimum of one player FT1 class needs to be competing in the field of play, and a maximum of one player FT3 class is allowed during the game. Moreover, these sports classes contain para-athletes with higher (i.e., FT1) and lower levels of impairment (i.e., FT3), respectively (Reina et al., 2021). These classification rules constrain the participation of players of those profiles and the availability of the number generating difficulties in the recruitment for studies. Possibly, the grouping according to playing position could be more feasible. According to the reviewer's comments, the following sentence has been included in the conclusions section as follows (Lines 388‒393 and 393‒399):

“Further studies could also consider larger international-level sample sizes to compare sports performance according to playing position and/or sport classes, in combination with the addition of multiple variables that are relevant to neuromuscular fatigue and the time-course recovery that can allow the design of individualized training programs approaching the competitive season.”

Also, more research is needed to explore the use of methods differentiating between respiratory and muscular perceived effort, alternatives that could provide an individual approach to quantify the internal load in para-footballers with CP [30]. In agreement, future research would examine differences in the match RPE between CP football players and non-impaired footballers, hypothesizing that data may provide insights to understand the feasibility of the use of tools for the subjective perception of the efforts in the specific para-sport contexts [53].”

  1. Runciman, P.; Tucker, R.; Ferreira, S.; Albertus-Kajee, Y.; Derman, W. Paralympic Athletes with Cerebral Palsy Display Altered Pacing Strategies in Distance-Deceived Shuttle Running Trials. Scand. J. Med. Sci. Sport. 2016, 26, 1239–1248, doi:10.1111/sms.12575.

Reference

Reina, R.; Iturricastillo, A.; Castillo, D.; Roldan, A.; Toledo, C.; Yanci, J. Is impaired coordination related to match physical load in footballers with cerebral palsy of different sport classes? J Sports Sci. 2021, 39(sup1), 140‒149. doi:10.1080/02640414.2021.1880740.

#3: The findings are significant and especially the difference between the post match difference in SJ & CMJ vs VJ performance is interesting and great to have published.

AA#3: Thank you for this positive feedback.

#4: great detailed discussion also highlighting findings of previously published material and answering some before the discussion unanswered questions regarding comparison to non cerebral palsy football players performance.

AA#4: Thank you for your comment.

#5: great and valid and important disclaimer again, at line 344. I look forward to seeing your article published. kind regards.

AA#5: Thank you for your feedback and contributions. We have found your criticism and recommendations very constructive.

Reviewer 3 Report

The authors have investigated perceived exertion in football in a specific group. Rating of perceived exertion is of course perceptual, and therefore, I am not sure why RPE would be different or unique? RPE would vary on how intense the exercise was and that would be intense if the person was not used to it, or the goal to be achieved was very important. That is, the exercise is worth doing. 

A limitation of the present study in both development of idea and discussion is that a strong rationale for what was done and why is not there. The information presented is useful but the authors dont guide the reader to know what its true value is; as  though to say.. here are the results, make from them what you want. As such, the article wont get the attention from readers the data might deserve. And so my recommendation is  that author revisit  the work and revise and re-write. I see these as major revisions. 

Author Response

Answers to Reviewer 3

General comments

Author's Answer (AA): We would like to thank the reviewer for the helpful advice and recommendations regarding this manuscript with ID ijerph-1145669. We revised the manuscript based on your suggestions on a point-by-point basis. In the following lines, we have included all the comments and clarifications for the reviewer's queries, informing you about the modifications in the new version of the manuscript (in red colour).

#1: The authors have investigated perceived exertion in football in a specific group. Rating of perceived exertion is of course perceptual, and therefore, I am not sure why RPE would be different or unique? RPE would vary on how intense the exercise was and that would be intense if the person was not used to it, or the goal to be achieved was very important. That is, the exercise is worth doing.

AA#1: Thank you for your comment and query. The reviewer is correct in declaring that the rating of perceived exertion (RPE) is based on a subjective perception of the effort, which is a response to the integration of sensations that could represent an oversimplification of the psychophysiological construct (Arcos et al., 2016). However, the RPE has been considered a valid and reliable method for quantifying match load in team sports (Azcárate et al., 2021; Arcos et al., 2016; Foster et al., 2001; Impellizzeri et al., 2004) and has also been used previously in CP football (Freitas et al., 2020; Henríquez et al., 2021). Previous studies reported that RPE responses could differ according to playing positions, varying due to the match physical demands and the specific role in the field of play (Azcárate et al., 2020). Based on these premises, it might be pertinent to hypothesize that RPE could vary according to sport classes due to the differences in the match's running performance (Yanci et al., 2018). This study demonstrates that CP football players with a national level described no differences in the RPE and perceived match load according to sport classes and field positions, possibly due to the specific characteristics of the para-sport (i.e., game rules, technical aspects, quality of opposition). To clarify the abovementioned ideas, the following information has been included in the new version of the manuscript (Lines 385‒388): 

A possible limitation of the present study is the no inclusion of other internal or external load variables which, in conjunction with the use of subjective perception tool and VJ, could provide a better comprehension of the load-fatigue process in competitive matches of CP football.

References

Arcos, A.L.; Méndez-Villanueva, A.; Yanci, J.; Martínez-Santos, R. Respiratory and muscular perceived exertion during official games in professional soccer players. Int. J. Sports Physiol. Perform. 2016, 11(3), 301–304. doi:10.1123/ijspp.2015-0270.

Azcárate, U.; Los Arcos, A.; Yanci, J. Variability of professional soccer players’ perceived match load after successive matches. Res. Sports Med. 2021, 29(4), 349–363. doi:10.1080/15438627.2020.1856104.

de Freitas, V.H.; Nakamura, F.Y.; Faria, F.R. de; Dantas, M.J. bezerra; Souza, N.C.; Buratti, J.R.; Freitas, A. de; Figueiredo, D.H.; Golçalves, H.R.; Junior, W.N.; Gorla, J.I. Internal training load and performance indices of cerebral palsy football players and effects of one week with and without training on heart rate variability. J. Phys. Educ. Sport. 2020, 20(5), 3017–3022. doi:10.7752/jpes.2020.s5410.

Foster, C.; Florhaug, J.A.; Franklin, J.; Gottschall, L.; Hrovatin, L.A.; Parker, S.; Doleshal, P.; Dodge, C. A new approach to monitoring exercise training. J. Strength Cond. Res. 2001, 15(1), 109–115.

Henríquez, M.; Iturricastillo, A.; González-Olguín, A.; Herrera, F.; Riquelme, S.; Reina, R. Time−motion characteristics and physiological responses of para-footballers with cerebral palsy in two small-sided games and a simulated game. Adapt. Phys. Act. Q. 2021, 1–16. doi:10.1123/apaq.2020-0077.

Impellizzeri, F.M.; Rampinini, E.; Coutts, A.J.; Sassi, A.; Marcora, S.M. Use of RPE-based training load in soccer. Med. Sci. Sports Exerc. 2004, 36(6), 1042–1047. doi:10.1249/01.mss.0000128199.23901.2f.

Yanci, J.; Castillo, D.; Iturricastillo, A.; Urbán, T.; Reina, R. External match loads of footballers with cerebral palsy: A comparison among sport classes. Int. J. Sports Physiol. Perform. 2018, 13(5), 590–596. doi:10.1123/ijspp.2017-0042.

#2: A limitation of the present study in both development of idea and discussion is that a strong rationale for what was done and why is not there. The information presented is useful but the authors dont guide the reader to know what its true value is; as though to say… here are the results, make from them what you want. As such, the article wont get the attention from readers the data might deserve. And so my recommendation is  that author revisit the work and revise and re-write. I see these as major revisions.

AA#2: Thank you for your comments and suggestions. Considering the reviewer's queries, we extended the information in different sections of the manuscript to improve the readability and consistency of this study:

Discussion

(Lines 286‒289): 

“Even though the present study doesn't find differences in the RPE and ML according to sports classes and player positions, the use of these methods could be helpful in a practical setting to perform a post-match individual monitorization considering the specific CP football characteristics.

(Lines 346‒349):  

The significant performance loss found in VJ after a competitive match should concern coaches for addressing adequate recovery periods during congested calendars or after intense games, especially for those who showed more jump height decrements such as FT3 players and those with MF or AT playing roles.

(Lines 377‒380):

The present data provide novel information suggesting that VJ tests could be a useful indirect measure of neuromuscular fatigue expressed in height jump loss after a football match, highlighting that CP football can be a modality with high neuromuscular demands.

(Lines 380‒382): 

“The SJ performance pointed out some differences regarding fatigue manifestation, considering players’ sport classes and playing positions, especially for less impaired players (i.e., FT3) and those assuming tactical roles in midfield and forward positions.

(Lines 382‒385):  

A practical application of using VJ tests is that technical staff could indirectly estimate neuromuscular and fatigue status, providing necessary information to adjust individual recovery strategies according to CP footballers’ requirements [14].”

Round 2

Reviewer 3 Report

The authors have provided comments in a letter defending their original decisions. However, as a reader, I would like a stronger justification of why perceived exertion is worth assessing. Just because other authors have done and published does not make it a correct decision for the present study. I would ask the authors to revisit the original comments and introduce arguments made in the rebuttal into the paper. I am unlikely to recommend publication until this has occurred.

Author Response

We appreciate the reviewers’ thoughtful comments, and we hope they will agree this updated version is much stronger as a result. In all areas covered by the reviewer, we have tried to deal with their concerns as fully as possible. Below we present a point-by-point overview of the changes we have made to the manuscript as requested by the reviewers. Those changes are highlighted in red colour in the manuscript file.

We appreciate your consideration, and we look forward to hearing from you soon.

Sincerely,

Authors

 Answers to Reviewer 3

Comments and Suggestions for Authors

The authors have provided comments in a letter defending their original decisions. However, as a reader, I would like a stronger justification of why perceived exertion is worth assessing. Just because other authors have done and published does not make it a correct decision for the present study. I would ask the authors to revisit the original comments and introduce arguments made in the rebuttal into the paper. I am unlikely to recommend publication until this has occurred.

Author's Answer (AA): Thank you for the new comments in this second round (R2) of this manuscript revision. These comments and modifications help us to improve our initial manuscript. We have added a rationale for the relevance of using a tool based on the subjective perception of effort in football players with cerebral palsy in the methods section. We agree that it is not sufficient that other studies have previously used it, so we have included information regarding the validity of this method of quantifying internal load in football matches. The following information has been included in the new version of the manuscript (Lines 118‒131): 

The rated perceived exertion (RPE) has increasingly been used to measure internal load and to determine the perceived effort in non-impaired football players [13,24,25,27] and previously utilized in players with CP [10,28]. This method has been described as valid for quantifying efforts during training and football matches [29,30,31] and has been proposed for use in the context of football and other team sports [32,33]. For example, Foster et al. [34,35,36] proposed the RPE to assess the `hardness´ of the entire training session and to evaluate internal load in endurance and team sport athletes. In addition, the RPE provides information in a simple, practical, and low-cost way [37]. Especially in high-level performance contexts, where the use of technology is constrained due to the discomfort when players are at peak performance in competition, the RPE can be an excellent non-invasive method applied once the match is over [36], minimizing the intrusion in the collection of information. Due to the validity of this method and its facility, versatility, and low cost, the RPE was used in this study to quantify the internal load of the players during the matches.

Due to the inclusión of this new piece of information in the manuscript, new references have been included (27, 29‒35 and 37), and the numbers of the references have been also modified in the methods and discussion sections.

Round 3

Reviewer 3 Report

Many thanks for revising the paper. My comments remain on the same issue of the justification for using perceived exertion to assess load. I wish the authors to continue this discussion and look at the limitations of using perceived exertion to assess load. There will be instances where perceived exertion might provide a valid measure of load, but other instances where it does not, where people over or under report RPE. This aspect of the work is very important as it provides a contextual limitation and therefore helps position the paper much better.

Author Response

Answers to Reviewer 3

Comments and Suggestions for Authors

Many thanks for revising the paper. My comments remain on the same issue of the justification for using perceived exertion to assess load. I wish the authors to continue this discussion and look at the limitations of using perceived exertion to assess load. There will be instances where perceived exertion might provide a valid measure of load, but other instances where it does not, where people over or under report RPE. This aspect of the work is very important as it provides a contextual limitation and therefore helps position the paper much better.

Author's Answer (AA): Thank you for the comments (R3). We agree with the reviewer. We have added the limitations of the RPE method to quantify match load at the end of the manuscript. In addition, we agree with the reviewer that this aspect may be relevant to readers.
